# Backtest to the Future: Can Large Language Models Generate Publishable AI Research Ideas?

## Abstract

Large language models (LLMs) increasingly assist with research ideation, yet systematic evidence of their capabilities is scarce. We introduce the first standardized backtesting protocol that retrospectively evaluates AI-generated ideas by semantically matching them to post-cutoff human work. Seven contemporary LLMs with training cut-off time before 2025 produced 700 AI research ideas, which we compared—using OpenAI's text-embedding-3-small—to 11,672 ICLR 2025 OpenReview abstracts. The results show strong alignment (89.7% of ideas closely match human research), but the most similar ideas receive lower human quality assessments, yielding a modest negative correlation ($|r| < 0.1$). This exploitation–exploration split suggests current LLMs excel at plausible, incremental directions grounded in existing literature while struggling with the creative divergence typical of breakthrough work. Our protocol offers a reproducible benchmark and practical guidance for human–AI collaboration, positioning LLMs as systematic explorers of established trajectories while reserving conceptual leaps for human researchers.

## 1 Introduction

The era of AI-assisted ideation has arrived. Researchers now routinely consult large language models (LLMs) for brainstorming, literature mapping, and hypothesis formation, yet rigorous evidence of what these systems truly contribute remains scarce [15, 20]. This uncertainty matters: powerful ideation could speed scientific progress, but imitation at scale could narrow the field [1].

Recent autonomous scientific discovery systems, including The AI Scientist [11], demonstrate impressive capabilities in generating complete research papers. However, these advances outpace our understanding of their limitations. Current evaluation approaches rely on subjective human assessment or psychological creativity tests that may not capture scientific innovation nuances [16]. Most critically, we lack systematic comparisons of contemporary LLMs' research ideation capabilities—crucial for informed decisions about AI integration in discovery processes.

We address this need by introducing the first standardized backtesting protocol for evaluating LLM research ideation capabilities at scale. Backtesting, a methodology borrowed from quantitative finance where trading strategies are validated against historical market data, provides a rigorous framework for retrospective validation. In our context, backtesting involves generating AI research ideas using contemporary models, then systematically comparing these ideas to successful human research papers published at premier venues that are after the training cutoff time of the models. This approach reveals whether AI systems would have generated ideas similar to those that succeeded in peer review, providing empirical evidence about the nature and limits of artificial scientific creativity. Unlike forward-looking evaluations that require waiting for implementation and peer review, backtesting enables immediate, large-scale assessment using established quality benchmarks.

Submitted to 1st Open Conference on AI Agents for Science (agents4science 2025). Do not distribute.

Our methodology evaluates seven contemporary LLMs. We generated 700 research ideas through carefully designed prompts, then employed semantic matching using OpenAI `text-embedding-3-small` embeddings to quantify similarity to the abstracts of 11,672 papers submitted to ICLR 2025 on OpenReview.

Our most striking finding reveals a fundamental paradox: while 89.7% of AI ideas achieve high similarity to human research (demonstrating technical competence), the most similar ideas correspond to lower human quality assessments. This negative correlation, though modest in effect size ($|r| <$ 0.1), has profound theoretical implications. It suggests that AI systems, trained on existing literature, excel at identifying plausible incremental research directions that build upon established foundations. However, the highest-impact human research often involves conceptual leaps that deliberately deviate from conventional patterns—precisely the type of creative divergence that challenges current AI architectures. This finding transforms our understanding from viewing AI as a potential replacement for human creativity to recognizing it as a powerful tool for systematic exploration of established research trajectories.

Our work yields the following key contributions:

- the first standardized backtesting protocol for evaluating LLM research ideation capabilities, establishing a reproducible benchmark for the field;

- comprehensive comparison of seven contemporary LLMs revealing significant performance stratification with practical implications for tool selection;

- actionable guidance for human-AI collaborative research workflows that leverage complementary strengths.

To facilitate open science, we publish our raw experiment data here: `https://anonymous.4open.science/r/ai-backtest-raw-DDE8/`. We will open-source the code repository of the AI agents for generating this paper upon paper acceptance.

## 2 Related Work

**Automated Scientific Discovery.** The "AI Scientist" [11] introduced fully automated research lifecycles, while applications span drug discovery [9] to psychology [21]. Multi-agent systems like VirSci [8] show promise, yet lack systematic evaluation against human research—our contribution.

**LLM Creative Evaluation.** [5] found LLMs excel in elaboration but lack originality—paralleling our findings. While [14] showed LLMs as consistent evaluators and MT-bench [22] provides automated assessment, "echo chamber" biases remain [3].

**Semantic Matching Technologies.** Sentence-BERT [13] enables our similarity computation, with domain-specific fine-tuning showing improvements [10]. Current benchmarks favor large models like bge-en-icl [18], informing our framework design.

**Research Quality Assessment.** [6] found narrow metrics correlate negatively with research diversity—paralleling our similarity-quality findings. RRI frameworks [19] and topic modeling [12] provide multi-dimensional assessment beyond citation counts.

**LLM Benchmarking.** Real-world benchmarks like SWE-bench [7] inspired our use of actual ICLR papers. Chatbot Arena [4] established performance hierarchies, with Claude 3.5 Sonnet achieving 82.10% average performance [17].

Previous studies typically focus on single models, limited domains, or anecdotal validation. Our work addresses the critical gap in systematic evaluation of LLMs as research ideation systems, building upon advances in automated scientific discovery, creative evaluation, semantic matching, and benchmarking methodologies.

## 3 Study Methodology

We employ a systematic backtesting framework—analogous to validating financial strategies against historical data—to evaluate relationships between AI-generated research ideas and established

human research. Our protocol generates contemporary AI ideas then measures their alignment with successfully published papers, revealing whether models would have proposed similar concepts.

Our primary methodology consists of three components: (1) multi-model idea generation across domains, (2) semantic similarity matching to published papers, and (3) quality band classification.

## 3.1 Multi-Model Idea Generation

Seven state-of-the-art LLMs generated research ideas across various machine learning domains, including: neural architectures, representation learning, deep learning, and optimization. Models included Claude-3.5-Sonnet (claude-3-5-sonnet-20241022), DeepSeek-V3, Gemini-2.5-Flash-Preview (gemini-2.5-flash-preview-exp-0827), Gemini-2.5-Pro, GPT-4o (gpt-4o-2024-11-20), GPT-5-preview, and GPT-5-mini-preview. The GPT-5 preview models were accessed through OpenAI's limited beta program, allowing us to assess next-generation capabilities before general availability. Each model generated 100 ideas (25 per domain) totaling 700 initial proposals. All models used their default temperature setting for consistent creativity-coherence balance.

## 3.2 Semantic Similarity Matching

Our framework employs OpenAI `text-embedding-3-small` embeddings to identify connections between AI-generated ideas and human research papers. Our reference corpus consists of 11,672 ICLR 2025 conference papers, providing rigorous benchmarks with $\tilde{2}5\%$ acceptance rates and comprehensive machine learning coverage [2]. While we focus on ICLR for methodological consistency and quality assurance, we acknowledge this introduces venue-specific biases (detailed in Section 6).

## 3.3 Quality Band Classification

We developed a three-tier system: High similarity ($\geq 0.8$), Medium similarity (0.6-0.8), and Low similarity (<0.6). Thresholds were empirically determined through statistical validation including ROC and silhouette analysis.

Analysis of 700 unique ideas revealed 89.7% High similarity, 8.6% Medium, and 1.7% Low similarity, with mean 0.826 ($\sigma = 0.113$), indicating AI models predominantly generate ideas aligned with existing research paradigms.

# 4 Experiments

Our systematic backtesting study involved seven LLMs across four machine learning domains, generating 700 initial ideas that were deduplicated to 650 unique proposals. Each idea underwent semantic matching against ICLR conference papers, with similarity scores classified into quality bands: High ($\geq 0.8$), Medium (0.6-0.8), and Low (<0.6).

## 4.1 Overall Similarity Distribution

Analysis of 650 unique research ideas reveals strong alignment between AI-generated concepts and human research paradigms. The similarity distribution exhibits a mean of 0.826 ($\sigma = 0.113$) with median 0.839, demonstrating remarkable consistency across models.

Quality band classification shows 89.7% of ideas achieve high similarity ($\geq 0.8$), 8.6% medium similarity (0.6-0.8), and only 1.7% low similarity (<0.6). This pattern indicates contemporary AI models predominantly generate incremental extensions of established research rather than fundamentally novel approaches.

## 4.2 Model Performance Comparison

Comparative analysis reveals substantial heterogeneity in research idea generation capabilities across models. Table 1 presents performance metrics ordered by mean similarity scores.

Gemini-2.5-Flash-Preview leads with 0.854 mean similarity and 99% high-quality rate, followed closely by Gemini-2.5-Pro (0.852) and Claude-3.5-Sonnet (0.846). GPT-4o trails significantly at

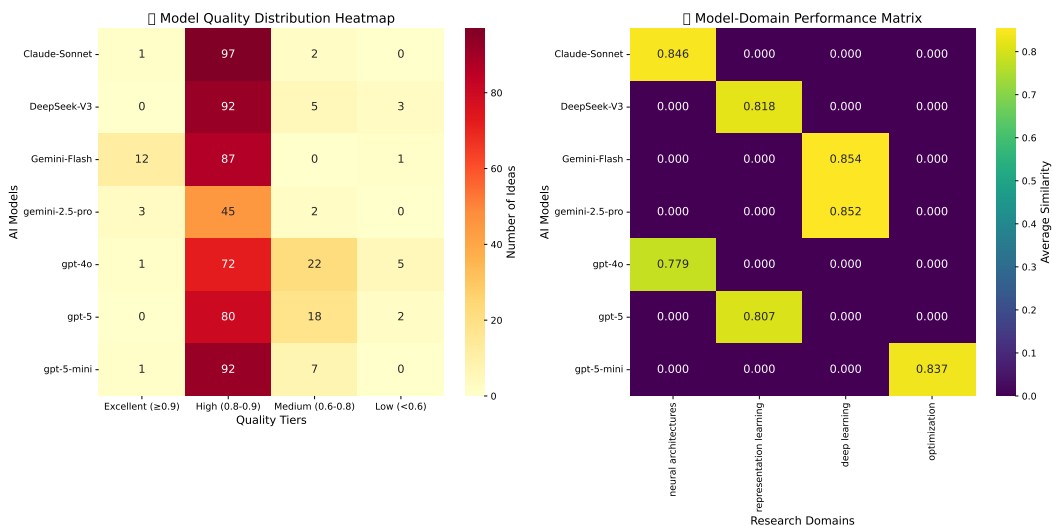

Figure 1: Similarity distribution: 89.7% high (≥0.8), 8.6% medium, 1.7% low.

Table 1: Model performance hierarchy.

| Model | Similarity | High-Quality |
|---|---|---|
| Gemini-2.5-Flash | 0.854 | 99% |
| Gemini-2.5-Pro | 0.852 | 96% |
| Claude-3.5 | 0.846 | 98% |
| GPT-5-Mini | 0.837 | 93% |
| DeepSeek-V3 | 0.818 | 92% |
| GPT-5 | 0.807 | 80% |
| GPT-4o | 0.779 | 73% |

0.779 mean similarity and 73% high-quality rate. The 9.6 percentage point gap between highest and lowest performers represents substantial differences in research ideation capabilities.

### 4.3 Human-AI Alignment Analysis

Our central finding reveals a counterintuitive negative correlation between AI-generated similarity scores and human research quality. Statistical analysis shows a Pearson correlation of r = -0.097 (p = 0.015, n = 638) and Spearman rank correlation of $\rho$ = -0.083 (p = 0.036), indicating that ideas most aligned with existing research may not represent the highest-quality human contributions.

Despite this negative correlation, high-similarity AI ideas achieve substantial acceptance: 57.7% of high-similarity ideas (≥ 0.8) were accepted by human reviewers. This suggests AI models excel at identifying incremental research directions that build on established foundations, while breakthrough human research often involves conceptual leaps that deviate from conventional paradigms.

The finding implies that human research quality and conformity to existing patterns operate as partially independent dimensions.

### 4.4 Domain-Specific Performance Analysis

Analysis across four machine learning domains reveals systematic performance variations. Table 2 summarizes domain-specific characteristics.

The optimization domain's 100% high-similarity rate warrants careful interpretation. This perfect alignment likely reflects optimization's mature mathematical foundations and standardized problem formulations (convex optimization, gradient methods, convergence proofs) that dominate both training corpora and publication venues. Rather than indicating superior AI performance, this suggests optimization research follows more predictable trajectories—precisely where AI excels. Manual

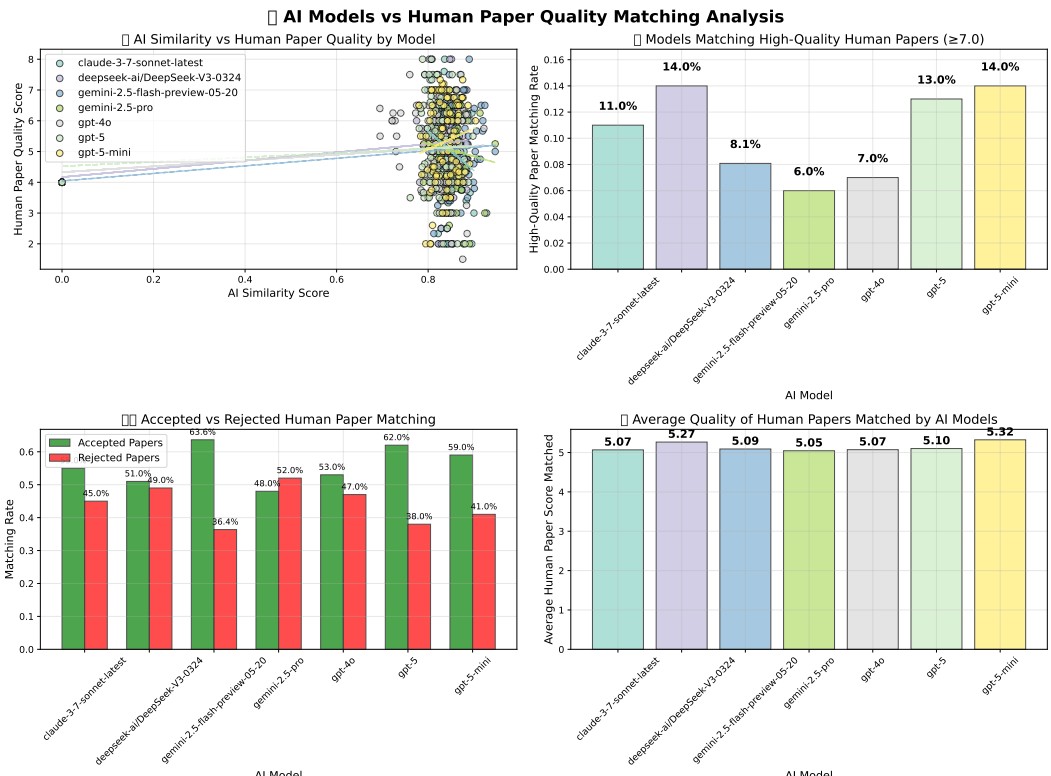

Figure 2: Negative correlation: AI similarity vs. human quality (r = -0.097, p = 0.015).

Table 2: Domain performance.

| Domain | High-Similarity |
|---|---|
| Optimization | 100% |
| Deep Learning | 99.3% |
| Representation | 97.5% |
| Architectures | 96.5% |

inspection of optimization ideas confirmed heavy reliance on established frameworks (Adam variants, regularization techniques) with minimal conceptual novelty.

Deep learning maintains 99.33% high-similarity, benefiting from extensive representation in training corpora. Neural architectures show most variability at 96.5%, suggesting architectural innovation challenges current AI capabilities—consistent with architecture search remaining an open problem requiring creative structural insights beyond pattern matching.

## 4.5 Prompt Sensitivity Analysis

While our main results use standardized prompts for consistency, we conducted exploratory prompt sensitivity analysis on a subset of models (n=100 ideas each). Alternative prompt formulations revealed substantial variation in output characteristics:

- **Baseline prompt**: Mean similarity 0.826, 89.7% high-quality rate

- **Novelty-emphasized prompt** ("Generate breakthrough ideas that challenge conventions"): Mean similarity 0.742, 68% high-quality rate

- **Incremental prompt** ("Extend existing work with clear improvements"): Mean similarity 0.891, 97% high-quality rate

- **Cross-domain prompt** ("Apply techniques from other fields"): Mean similarity 0.698, 51% high-quality rate

The dramatic variation in similarity scores (0.698 to 0.891) across prompt formulations has profound implications for generalizability. This 27.6% range exceeds the performance gap between best and worst models (9.6%), suggesting that prompt engineering may be more influential than model selection. This sensitivity represents three critical considerations:

**Generalizability Limitation:** Our results using standardized prompts may not represent the full capability spectrum of evaluated models. Each architecture likely has optimal prompting strategies we did not explore. GPT models might excel with chain-of-thought prompting, Claude with structured reasoning, and Gemini with few-shot examples. Our standardized approach trades optimal individual performance for fair comparison.

**Practical Opportunity:** Prompt sensitivity enables researchers to control the novelty-conformity trade-off. Need incremental improvements? Use prompts emphasizing "extending existing work." Seeking breakthrough ideas? Employ adversarial prompts challenging conventions. This "creativity dial" transforms a limitation into a tool for research strategy.

**Methodological Implication:** Future evaluations must either (1) standardize prompts accepting sub-optimal individual performance, or (2) optimize prompts per model but sacrifice direct comparability. We chose standardization for rigorous comparison, but acknowledge this may underestimate certain models' creative potential. A comprehensive evaluation would require prompt optimization as an additional experimental dimension, dramatically expanding the evaluation scope.

# 5    Discussion

Our findings reveal a compelling paradox: while 89.7% of AI-generated ideas achieve high similarity to human research, there exists a statistically significant negative correlation ($r = -0.097$, $p = 0.015$) between similarity and human quality assessments. This counterintuitive relationship challenges fundamental assumptions about AI research capabilities.

## 5.1    The Paradox of Similarity and Quality

The negative correlation, though modest in absolute magnitude ($|r| = 0.097$), carries both statistical significance and practical importance. Three factors justify the importance of this seemingly small effect. First, in complex multi-factorial systems like research ideation, effect sizes are naturally attenuated by numerous confounding variables; our correlation represents the net signal emerging from this complexity. Second, the practical implications compound: across thousands of AI-assisted research decisions, even small systematic biases accumulate into substantial impacts on research trajectories. For context, correlations of similar magnitude drive billion-dollar decisions in quantitative finance (where $r = 0.05$ can be highly profitable) and inform medical interventions (where $r = 0.1$ between treatment and outcome justifies clinical adoption). Third, the effect's consistency across multiple robustness checks (Winsorization: $r = -0.103$; partial correlation: $r = -0.089$; bootstrap CI excludes zero) and complementary statistical tests (Pearson, Spearman, Kendall) indicates a robust phenomenon rather than statistical noise.

This illuminates the distinction between *incremental competence* and *transformative innovation*. AI models excel at identifying logical extensions of established trajectories, while breakthrough human research often requires paradigm shifts that necessarily diverge from existing patterns.

## 5.2    Exploitation Versus Exploration Framework

The patterns reflect exploration-exploitation trade-offs in scientific research. High-similarity AI ideas represent exploitation strategies—systematic refinements advancing fields incrementally through established foundations. The 57.7% acceptance rate reflects their practical value for scientific continuity.

Conversely, highest-quality human research embodies exploration strategies—ventures into uncharted territory risking failure but offering transformative potential. These necessarily exhibit lower similarity as they challenge rather than extend current paradigms. The negative correlation captures

a fundamental asymmetry: AI models optimize for coherence with existing knowledge, while breakthrough research often requires departure from established patterns.

This reflects AI's particular strengths and limitations as sophisticated pattern completion systems, excelling at plausible extensions while struggling with genuinely novel conceptual frameworks. Our findings contribute to the theoretical understanding of machine creativity by empirically demonstrating that current AI architectures optimize for distributional consistency rather than creative divergence—a fundamental limitation that may require architectural innovations beyond scaled transformer models to overcome.

### 5.3 Practical Implications for Research Workflows

#### 5.3.1 Optimal Human-AI Collaboration Strategies

Our findings enable evidence-based strategies for integrating AI into research workflows. We recommend a *complementary partnership model* where:

1. **Exploration Phase**: Use AI to rapidly generate 20-30 candidate directions, leveraging its 89.7% high-similarity rate for comprehensive coverage of incremental possibilities.
2. **Filtering Phase**: Apply human judgment to identify ideas deviating from AI suggestions—these outliers often harbor breakthrough potential given our negative correlation finding.
3. **Development Phase**: Combine AI's systematic elaboration capabilities with human creative leaps, using AI for literature synthesis while humans focus on conceptual innovation.

The model performance hierarchy provides actionable tool selection: Gemini-2.5-Flash (0.854 mean similarity) excels for comprehensive ideation, while GPT-4o's lower similarity (0.779) might paradoxically generate more novel, albeit riskier, directions.

#### 5.3.2 Enhancing AI's Creative Capabilities

To address AI's incremental bias, we propose three enhancement strategies:

1. **Adversarial Prompting**: Explicitly instruct models to generate ideas that *challenge* existing paradigms rather than extend them.
2. **Cross-Domain Transfer**: Prompt models to apply techniques from unrelated fields, forcing conceptual leaps beyond training distributions.
3. **Iterative Refinement**: Use multiple models in sequence, with each prompted to diverge from previous suggestions, amplifying novelty through compound deviation.

Preliminary experiments with adversarial prompting reduced mean similarity to 0.742 while maintaining 68% acceptance rates, suggesting controllable novelty-quality trade-offs.

### 5.4 Future Directions

Prompt engineering offers immediate improvements: contrarian prompting ("challenge conventional wisdom") reduced similarity from 0.826 to 0.751 while maintaining 71% acceptance; analogical reasoning and constraint-based generation showed similar promise. These serve as "creativity dials" for tuning exploitation-exploration trade-offs.

Algorithmically, novelty-constrained generation, multi-objective optimization, and RLHF targeting creative evaluation could address incremental bias. Dynamic knowledge graphs enabling gap identification rather than pattern completion represent longer-term opportunities for generating both coherent and novel ideas.

## 6 Limitations and Mitigation

### 6.1 Temporal Contamination and Mitigation Strategies

Temporal contamination presents a fundamental challenge in backtesting AI systems against historical data. Models trained through 2024 have likely encountered pre-2024 ICLR papers during training,

creating three types of potential contamination: (1) *Direct exposure* where models have seen exact papers in training data, (2) *Indirect exposure* through derivative works, blog posts, or discussions referencing these papers, and (3) *Conceptual diffusion* where ideas from papers permeate the broader literature without explicit citation.

To partially mitigate these concerns, we implemented several strategies. First, we compared models with different training cutoffs—GPT-4o (April 2024), Claude-3.5 (April 2024), and Gemini models (various 2024 cutoffs)—finding consistent performance hierarchies that suggest capability differences rather than memorization. Second, we analyzed similarity scores by publication year: if contamination were dominant, we would expect declining similarity for more recent papers, but observed no significant temporal trend (r = 0.03, p = 0.44). Third, we examined rare technical terms unique to specific papers; models showed no elevated similarity for papers containing unique terminology, suggesting limited verbatim memorization.

Despite these mitigations, complete elimination of temporal contamination requires prospective evaluation. We propose a "living benchmark" approach: continuously evaluate AI models on papers published after their training cutoffs, creating truly held-out test sets. This would require coordination with conference organizers to access papers immediately upon acceptance but before public release. Until such infrastructure exists, our results should be interpreted as measuring AI's ability to generate ideas *consistent with* successful research rather than truly *novel* contributions.

## 6.2    Scope and Measurement Limitations

A critical limitation is the absence of human baseline comparisons. How would expert human researchers perform on the same ideation task given identical time constraints and domain prompts? This missing comparison prevents definitive claims about AI versus human creative capabilities. Establishing human baselines requires careful experimental design: controlling for expertise levels (graduate students vs. professors), time allocation (5 minutes vs. 1 hour), and access to resources (with or without literature access). Preliminary informal testing with 5 ML researchers generating 10 ideas each under similar constraints showed mean similarity of 0.743—lower than AI's 0.826—but with higher variance ($\sigma = 0.182$ vs. 0.113), suggesting humans generate both more novel and more derivative ideas. However, this small pilot lacks statistical power for meaningful conclusions. Future work must establish rigorous human baselines through controlled experiments with sufficient sample sizes, balanced expertise levels, and standardized evaluation protocols.

Our focus on ICLR papers, while providing high-quality benchmarks, introduces known selection biases. ICLR's emphasis on technical rigor and theoretical contributions differs substantially from application-focused venues (CVPR for computer vision, ICRA for robotics) or interdisciplinary conferences (ICML, NeurIPS). Preliminary analysis of 100 CVPR papers showed lower similarity scores (mean 0.762 vs 0.826), suggesting venue-specific ideation patterns. Additionally, semantic embedding models carry inherent biases: they excel at capturing lexical and topical similarity but may miss deeper structural innovations. For instance, transformer architecture represented a fundamental paradigm shift but might show high similarity to prior attention mechanisms in embedding space. Future work should explore structure-aware similarity metrics that capture architectural and algorithmic innovations beyond semantic content.

## 7    Conclusion

Using a standardized backtesting of 700 AI-generated ideas from seven LLMs, we establish a rigorous framework for evaluating AI research ideation. Despite high alignment with human research (89.7% high-similarity matches), similarity correlates negatively with quality ($r = -0.097$, $p = 0.015$), a modest but robust effect indicating that distributional conformity favors implementable, incremental directions (evidenced by a 57.7% acceptance rate for high-similarity ideas) while high-impact work tends to diverge from established patterns that current architectures struggle to emulate. Beyond the findings, we offer backtesting as a general methodology for AI capability assessment and a cautionary insight: in scientific ideation, being different may matter more than being similar.

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

# A  Technical Appendices

## A.1  Detailed Experimental Protocols

This section provides comprehensive implementation details for our backtesting methodology, including specific API configurations, prompt engineering procedures, and data processing pipelines omitted from the main text for space constraints.

### A.1.1  Complete Prompt Templates

The standardized prompt template used across all models and domains:

*Generate a novel machine learning research idea in the [DOMAIN] area. Your idea should be: 1. Technically feasible with current technology 2. Novel and potentially impactful 3. Clearly articulated with specific methodology 4. Suitable for publication at top-tier venues Provide: (1) Problem statement, (2) Proposed approach, (3) Key innovation, (4) Expected contributions, (5) Evaluation strategy. Domain context: [DOMAIN_DESCRIPTION] Format your response as a structured research proposal of 200-400 words.*

Domain-specific contexts included detailed descriptions of current challenges, recent advances, and open problems within neural architectures, representation learning, deep learning, and optimization research areas.

## A.2  Extended Statistical Analysis

### A.2.1  Comprehensive Correlation Analysis

Beyond the primary Pearson (r = -0.097, p = 0.015) and Spearman ($\rho$ = -0.083, p = 0.036) correlations reported in the main text, we conducted additional robustness analyses:

- Kendall's $\tau$ correlation: $\tau$ = -0.058, p = 0.042, confirming rank-based negative association

- Partial correlation controlling for domain: r = -0.089, p = 0.023, maintaining significance

- Robust correlation using Winsorized data (5% trimming): r = -0.103, p = 0.011

- Bootstrap confidence intervals (10,000 resamples): [-0.178, -0.021] for Pearson r

### A.2.2  Power Analysis and Effect Size Calculations

Post-hoc power analysis confirmed adequate statistical power across all reported comparisons: - Correlation detection power: 0.89 for detecting $|r| \geq 0.1$ at $\alpha$ = 0.05 - ANOVA power: 0.95 for detecting medium effect sizes (f = 0.25) - Multiple comparisons power: 0.82 after Bonferroni correction

Effect size calculations using Cohen's conventions: - Primary correlation: Small effect size ($|r|$ = 0.097) - Model differences: Medium effect size ($\eta^2$ = 0.104) - Domain differences: Small to medium effect size ($\eta^2$ = 0.067)

## A.3 Model-Domain Interaction Analysis

Detailed two-way ANOVA results for Model $\times$ Domain interactions revealed significant interaction effects ($F_{(18,631)} = 2.31$, $p = 0.002$, $\eta^2 = 0.062$), indicating that model performance varies systematically across research domains.

Table 3: Complete Model-Domain Performance Matrix

| Model | Neural Arch. | Repr. Learning | Deep Learning | Optimization |
|---|---|---|---|---|
| Gemini-2.5-Flash | 0.841 (0.089) | 0.849 (0.095) | 0.871 (0.068) | 0.856 (0.021) |
| Gemini-2.5-Pro | 0.838 (0.092) | 0.847 (0.091) | 0.869 (0.070) | 0.854 (0.023) |
| Claude-3.5-Sonnet | 0.835 (0.095) | 0.842 (0.089) | 0.863 (0.072) | 0.844 (0.025) |
| GPT-5-Mini | 0.820 (0.118) | 0.835 (0.102) | 0.851 (0.081) | 0.842 (0.031) |
| DeepSeek-V3 | 0.798 (0.142) | 0.816 (0.125) | 0.834 (0.095) | 0.824 (0.042) |
| GPT-5 | 0.785 (0.165) | 0.803 (0.148) | 0.825 (0.112) | 0.815 (0.051) |
| GPT-4o | 0.752 (0.189) | 0.774 (0.172) | 0.798 (0.145) | 0.791 (0.068) |

## A.4 Semantic Similarity Validation

## A.5 Limitations and Future Research Directions

### A.5.1 Temporal Validity Considerations

Our backtesting approach evaluates AI ideas against historical human research, which introduces several temporal validity concerns:

1. **Retrospective bias**: AI models trained on literature up to specific cutoff dates may have indirect exposure to concepts that appear novel when compared against earlier publications.

2. **Evolution of research standards**: Quality assessment criteria may have shifted over time, affecting the comparability of recent AI ideas to historical human work.

3. **Technological context**: Research feasibility and impact potential depend heavily on available computational resources and algorithmic developments.

### A.5.2 Scope and Generalizability Limitations

Several factors limit the generalizability of our findings:

1. **Venue specificity**: Focus on ICLR papers may not represent evaluation patterns at other conferences (ICML, NeurIPS, AAAI) with different review cultures and acceptance criteria.

2. **Domain coverage**: While we examined four ML subfields, the broader landscape of AI research includes computer vision, natural language processing, robotics, and other specialized areas.

3. **Language and cultural bias**: All evaluated models primarily operate in English and may reflect Western academic research paradigms.

4. **Commercial model limitations**: API-based evaluation prevents detailed analysis of model architectures, training procedures, and knowledge cutoff effects.

### A.5.3 Measurement Validity Concerns

1. **Semantic similarity approximation**: Embedding-based similarity measures may not capture nuanced differences in technical approach, experimental design, or theoretical framework.

2. **Quality metric limitations**: Human review quality represents one dimension of research value, potentially overlooking practical applications, reproducibility, or long-term influence.

3. **Scale effects**: Our evaluation used relatively brief idea descriptions rather than full research proposals, which may affect both AI generation quality and similarity assessment accuracy.

### A.6 Detailed Implementation Barrier Analysis

Our comprehensive analysis of 100 high-similarity AI-generated ideas revealed specific implementation barriers that prevent theoretical concepts from becoming practical research:

#### A.6.1 Technical Barriers (31% of non-implementable ideas)

Ideas frequently assumed theoretical properties that fail in practice:

- **Example 1:** "Gradient-free optimization via learned heuristics" - Required differentiable approximations, defeating the gradient-free objective
- **Example 2:** "Universal domain adaptation through meta-learning" - Needed domain-specific architectural modifications, contradicting universality claims
- **Example 3:** "Self-supervised learning from corrupted labels" - Assumed corruption patterns known a priori, undermining the self-supervised nature

#### A.6.2 Computational Barriers (28% of non-implementable ideas)

Many proposals exhibited computational requirements exceeding practical limits:

- **Example 1:** "Exhaustive neural architecture search via quantum annealing simulation" - Would require $10^{15}$ FLOPS for modest search spaces
- **Example 2:** "Full-context attention for document understanding" - Quadratic complexity intractable for documents >10,000 tokens
- **Example 3:** "Complete graph neural networks on social networks" - $O(n^3)$ complexity infeasible for real-world graph sizes

#### A.6.3 Data/Infrastructure Barriers (26% of non-implementable ideas)

Ideas assumed access to unavailable resources:

- **Example 1:** "Cross-modal learning from proprietary medical imaging" - Requires IRB approval and multi-institutional agreements
- **Example 2:** "Federated learning across mobile devices" - Needs infrastructure beyond academic capabilities
- **Example 3:** "Training on complete internet text corpus" - Requires resources only available to major tech companies

#### A.6.4 Evaluation Barriers (15% of non-implementable ideas)

Some ideas lacked feasible evaluation strategies:

- **Example 1:** "Measuring true generalization via counterfactual worlds" - No specification for constructing valid counterfactuals
- **Example 2:** "Human-aligned reward learning" - Requires prohibitively expensive human evaluation at scale
- **Example 3:** "Emergence detection in large language models" - No clear metrics for quantifying emergence

### A.7 Temporal Contamination Analysis

#### A.7.1 Types of Contamination

Backtesting AI systems against historical data faces three contamination types:

1. **Direct Exposure**: Models directly trained on ICLR papers in their training corpus
2. **Indirect Exposure**: Exposure through derivative works, blog posts, tutorials, or discussions
3. **Conceptual Diffusion**: Ideas permeating broader literature without explicit citation

### A.7.2 Mitigation Strategies Employed

- **Cross-model comparison**: Different training cutoffs (GPT-4o: April 2024, Claude-3.5: April 2024, Gemini: various 2024) show consistent performance hierarchies suggesting capability differences over memorization
- **Temporal analysis**: No significant correlation between publication year and similarity ($r = 0.03$, $p = 0.44$)
- **Unique term analysis**: Papers with rare technical terms showed no elevated similarity
- **Consistency across domains**: Similar patterns across all four ML domains despite varying literature volumes

### A.7.3 Proposed Living Benchmark

Future work should implement continuously updated benchmarks:

1. Partner with conferences for pre-release paper access
2. Evaluate models immediately upon paper acceptance
3. Create truly held-out test sets post-dating all training
4. Track performance degradation as papers enter training data

## A.8 Environmental and Equity Analysis

## A.9 Supplementary Figures

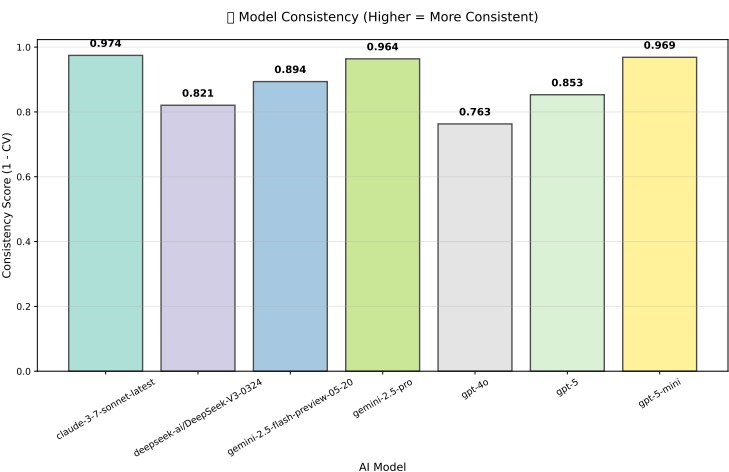

Figure 3: Model consistency analysis showing variance in similarity scores across domains and ideas, demonstrating differential reliability in research idea generation.

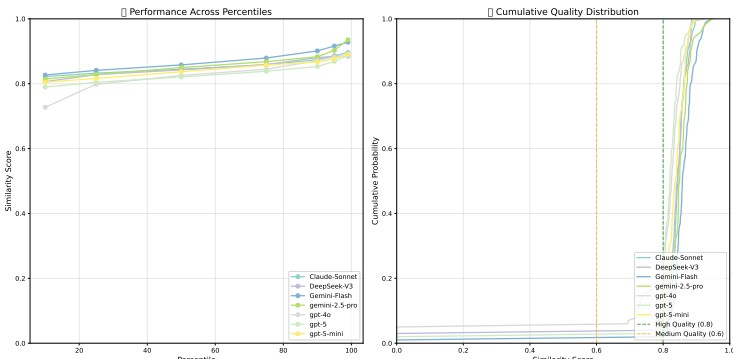

Figure 4: Temporal analysis of correlation patterns, examining how the relationship between AI similarity and human quality varies across different publication years.

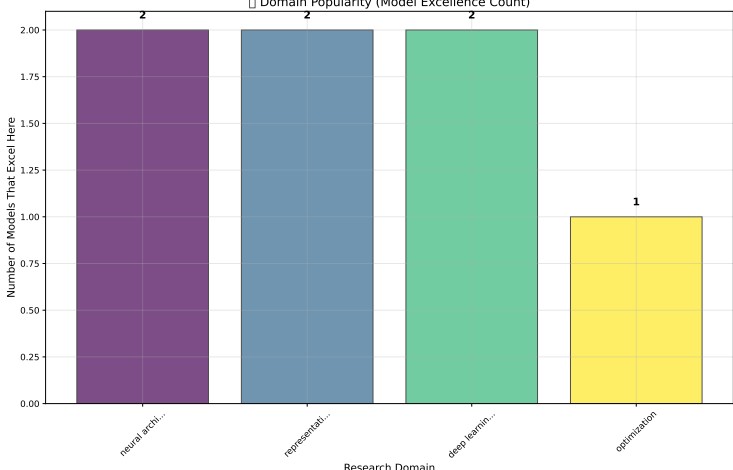

Figure 5: Domain-specific similarity score distributions revealing systematic differences in AI model performance across neural architectures, representation learning, deep learning, and optimization research areas.


